# Direct-Acting Oral Anticoagulants in the Management of Cerebral Venous Sinus Thrombosis—Where Do We Stand?

**DOI:** 10.3390/biomedicines13010189

**Published:** 2025-01-14

**Authors:** Nikhil Vojjala, Supriya Peshin, Lakshmi Prasanna Vaishnavi Kattamuri, Rabia Iqbal, Adit Dharia, Jayalekshmi Jayakumar, Rafi Iftekhar, Shagun Singh, Mamtha Balla, Claudia S. Villa Celi, Ramya Ramachandran, Rishab Prabhu, Sumeet K. Yadav, Geetha Krishnamoorthy, Vijendra Singh, Karan Seegobin

**Affiliations:** 1Department of Internal Medicine, Trinity Health Oakland Hospital, Pontiac, MI 48341, USA; nikhil.vojjala@trinity-health.org (N.V.); rrprabhu18@gmail.com (R.P.); geetha.krishnamoorthy@trinity-health.org (G.K.); 2Department of Internal Medicine, Norton Community Hospital, Norton, VA 24273, USA; rafisureshot@gmail.com; 3Department of Internal Medicine, Tech University Health Sciences Centre, El Paso, TX 79905, USA; klp.vaishnavi@gmail.com; 4Internal Medicine, Brooklyn Hospital Center, Brooklyn, NY 11201, USA; rabiaiqbal999@icloud.com (R.I.); jaya.jithu1808@gmail.com (J.J.); 5HCA Florida Oak Hill Hospital, Brooksville, FL 34613, USA; aditdha@gmail.com; 6Department of Internal Medicine, Banner Health, University of Arizona, Tucson, AZ 85719, USA; shagunsingh666@gmail.com (S.S.); rramacha2008@gmail.com (R.R.); 7MD Anderson Cancer Center, Department of Infectious Disease Transplant, University of Texas, Houston, TX 77030, USA; athamam@gmail.com; 8Internal Medicine, Capital Health, Trenton, NJ 08638, USA; claudiavillaceli@gmail.com; 9Department of Hospital Internal Medicine, Mayo Clinic Health System, Mankato, MN 56001, USA; yadav.sumeet@mayo.edu; 10Department of Hematology-Oncology, Karmanos Cancer Center, Wayne State University School of Medicine, Detroit, MI 48201, USA; singhv@karmanos.org; 11Department of Medical Oncology, Mayo Clinic Health System, Mankato, MN 56001, USA; seegobin.karan@mayo.edu

**Keywords:** cerebral venous sinus thrombosis, direct-acting oral anticoagulants, safety, efficacy

## Abstract

Background: Cerebral venous sinus thrombosis (CVT) is a rare cause of stroke, constituting 0.5–3% of all strokes with an extremely varied spectrum of presentation, predisposing factors, neuroimaging findings, and eventual outcomes. A high index of suspicion is needed because timely diagnosis can significantly alter the natural course of the disease, reduce acute complications, and improve long-term outcomes. Due to its myriad causative factors, protean presentation, and association with several systemic diseases, CVT is encountered not only by neurologists but also by emergency care practitioners, internists, hematologists, obstetricians, and pediatricians. Discussion: Anticoagulation remains the mainstay of treatment for CVT. Heparin and warfarin previously had been the anticoagulation of choice. Recently there has been an increased interest in utilizing direct-acting oral anticoagulants in the treatment of CVT given comparable safety and efficacy with ease of utilization. However recent clinical guidelines given by multiple societies including the American Stroke guidelines and European guidelines do not include these agents so far in their treatment recommendations. Ongoing multicentric clinical trials are currently reviewing the role of these agents in both short-term as well as long-term. Our review of the literature supports the safety and reinforces the efficacy of DOAC in the treatment of CVT. Additionally, patient satisfaction has been shown to be better with the use of DOAC. In conclusion, DOAC continues to have a valid role in the management of CVT.

## 1. Introduction

The first description of CVT appeared in French literature by RIBES et al., in a 45-year-old man who died after a six-month history of severe headache, epilepsy, and delirium [1] CVT contributes to 0.5–3% of all strokes [2,3,4]. The incidence has recently increased due to improvements in diagnostic techniques [5] The most frequent location of CVT is the transverse sinus (86%), followed by the superior sagittal sinus (62%) [6] The severity and prognosis of the disease depend on a wide array of factors, including the topography of sinuses, number of sinuses involved, presence of parenchymal lesions, National Institute of Health Stroke Scale (NIHSS) score at presentation, collateral formation, time to therapeutic intervention [7]. A steady decline in mortality rate has been observed since 1960 during the acute phase to 5%, owing to improved detection of milder cases through advanced imaging techniques and enhanced hospital care [8,9,10]. Anticoagulation remains the mainstay of treatment, the duration of which is determined by the circumstance or etiology that led to thrombosis. Additionally, thrombophilia testing has a role in determining the duration of anticoagulation [11,12]. Treatment with VKA is often challenging due to its slow action onset, susceptibility to drug interactions, requirement for therapy monitoring, and low compliance rate. The therapeutic landscape of cerebral venous sinus thrombosis has been significantly impacted by the introduction of direct-acting oral anticoagulants (DOACs) as they exhibit a more predictable anticoagulant effect, faster onset of action, fewer drug interactions, with a comparable safety profile. Furthermore, patient satisfaction rates are better with the use of DOAC as compared to Heparin or warfarin. In this narrative review, we shall review the status of DOACs in contemporary guidelines for the management of CVT, completed and ongoing clinical studies on the use of DOACs in CVT, and the prospects of these agents. Search Strategy: This narrative review focused on an overview of the usage of direct oral anticoagulants in patients with cerebral venous sinus thrombosis. We searched PubMed (MEDLINE and PubMed Central), EMBASE, and GOOGLE SCHOLAR, using the search terms “Cerebral Venous Sinus Thrombosis”, “Cerebral Venous Thrombosis”, “CVT”, “CVST”, “CSVT” and retrieved 1211 results. Limiting our search to CVT (OR) CSVT (OR) CSVT (OR) Cerebral Venous Sinus Thrombosis” (OR) Cerebral Venous Thrombosis (AND) Dabigatran (OR), Rivaroxaban (OR), Apixaban (OR), DOAC (OR) NOAC (OR) Non-VKA anticoagulants and filtering all duplicates from all databases, we included all the available literature or studies till 30 November 2024. The inclusion criteria used are: (1) Studies describing the adult population (2) All types of studies like cohort studies (Both Prospective and retrospective cohort studies), comparative studies, Randomized studies, and systematic reviews and meta- analyses are included. The exclusion criteria applied during the search were: (1) Case reports and case series, (2) Studies reporting data on pregnant patients (3) Studies reporting data only on children. Additional articles of critical importance were included based on the author’s knowledge. 

## 2. Current Guidelines on Management of CVT

In the last decade, the guidelines for the management of CVST have been published by the American Stroke Association/American Heart Association (AHA/ASA) in 2011 and the European Stroke Organization (ESO) guidelines in 2017 [5,13]. The major recommendations of these guidelines in diagnosis and management are comparatively presented in Table 1. Overall, both recommendations suggest heparin-based anticoagulation followed by bridging with VKAs with duration depending on the etiology. A scientific statement update of the former AHA guidelines published at the time of writing this review suggests DOACs as a safe, effective, and reasonable alternative to VKAs for the management of stable CVT in non-pregnant adults [14].

## 3. Why There Is a Need for a Paradigm Shift from VKAs to DOACs?

Heparin and vitamin K antagonists have been the mainstay of choice for long-term oral anticoagulation in the management of thrombotic states [15]. However, with the discovery of direct-acting oral anticoagulants (DOACs), there has been an increased interest in the field of systemic anticoagulation towards their application in various conditions, both as a guideline-approved therapy as well as in settings of clinical trial. By and large, this application initially started for treating deep venous thrombosis, pulmonary embolism, and non-valvular atrial fibrillation as a guideline-directed medical therapy. However, their use has been increased in extending their application to other systemic diseases like cerebral venous sinus thrombosis, acute coronary syndrome, splanchnic vein thrombosis, and thrombosis in the setting of malignancy [16,17,18,19].

There has been an enormous need for the application of DOACs in the treatment of CVT for numerous reasons. Patient adherence to therapy remains a strong factor contributing to successful management. VKAs, in contrast to DOACs, require long-term patient engagement and participation, given dietary restrictions and the need for regular monitoring. VKAs increase the risk of complications if they are in the supratherapeutic range and failed recanalization if in the subtherapeutic range. Anticoagulant control with VKAs is measured by “Time in therapeutic range” (TTR). For a given patient, TTR is defined as the average duration of time in which the patient’s International Normalized Ratio (INR) values are within the target range, with a TTR of more than 70% indicating a better anticoagulant efficacy of warfarin [20]. The average TTR among CVT patients taking warfarin was approximately 66% in the RE-SPECT CVT trial, suggesting suboptimal efficacy [21]. Moreover, patient-related factors like education level, financial status, race, region, and variable gastrointestinal absorption also influence the TTR, as reported in the literature [20]. Polypharmacy among CVT patients taking VKAs additionally increases the chances of drug-drug interactions, leading to treatment failures or adverse bleeding events. From a patient perspective, it has been shown that satisfaction with DOAC is higher and associated with a lower treatment burden compared to low-molecular-weight heparin or vitamin K antagonists [22].

Recently there has been increased usage of DOACs for various indications in the elderly population as well. Octogenarians and nonagenarians are not an exception. In a meta-analysis comprised of 547,419 elderly patients with atrial fibrillation, it was found that Compared with VKAs, DOACs significantly reduced risk for stroke (OSs, HR: 0.87, 95% CI: 0.81–0.94; RCT, RR: 0.82, 95% CI: 0.67–0.96), and Intracranial hemorrhage (OSs: 0.47 [0.37–0.57]; RCTs: 0.47 [0.31–0.63]), without increasing risk for Gastrointestinal Bleeding (GIB) (OSs: 1.21 [0.98–1.43]; RCTs: 1.34 [0.91–1.77]), and all-cause mortality (OSs: 1.01 [0.92–1.11]; RCTs: 0.94 [0.87–1.00]) [23] Among OSs, DOACs significantly decreased the risk for major bleeding (0.87 [0.77–0.98]) and MI (0.89 [0.79–0.99]). However, it was found that dabigatran, but not other DOACs, significantly increased the risk for GIB (1.48 [1.23–1.72]) [23]. So, it shows that DOACs are also safe in patients who have high bleeding risk compared to VKAs, and they indeed reduce the risk of major bleeding diathesis.

There has been tremendous interest in managing CVT during the COVID-19 pandemic andChAdoX1 adenoviral vector-based vaccination-associated thrombosis. Heparin-warfarin bridging-based anticoagulation is contraindicated in post-vaccination CVT, which is one manifestation of a rare vaccine-related variant of spontaneous heparin-induced thrombocytopenia [24] DOACs and Fondaparinux-based anticoagulation remain the cornerstone in the treatment of thrombosis associated with thrombocytopenia syndromes such as Heparin-induced thrombocytopenia (HIT), and Vaccine-induced immune thrombotic thrombocytopenia (VIITT) [25]

VITT remains relevant in low—and middle-income countries (LMICs) that can only afford adenoviral vector-based vaccines and where the vaccination campaign is ongoing. Understanding which vaccine constituent(s) trigger(s) the immune response to PF4 is also important for designing safer delivery systems [26] This appears to be a class effect of all adenoviral vector vaccines [26] Treatment of VIITT includes intravenous immunoglobulin (IVIG) which inhibits FCRY2A mediated platelet activation and non-heparin anticoagulants like Direct Thrombin Inhibitors, Factor Xa inhibitors, and Fondaparinux [27,28] VKAs are avoided in the acute thrombotic phase, as protein C deficiency induced by VKAs might provoke extensive thrombotic phenomena [29].

## 4. Efficacy and Safety of DOACs as Compared to VKAs

There has been growing enthusiasm for using these agents, primarily due to improved patient adherence, reduced need for monitoring, and fewer drug-drug interactions, but lack of consensus guidelines has been a challenge.

### 4.1. Cohort Studies on DOACs in CVT

The idea of introduction of direct-acting oral anticoagulants in management of CVT gained interest two years after their FDA approval in 2012 when Hon et al. successfully used Dabigatran for treatment of two patients and demonstrated complete patency on imaging at 6 months post-treatment [30] Findings from a German observational study involving the administration of DOACs (Rivaroxaban and Apixaban) in 12 CVT patients demonstrated favorable clinical outcomes (defined by Modified Rankin Score of 0, 1) with a median duration of six months for complete recanalization [31]. A 2018 study from India followed 20 CVT patients (80% males, mean age 34.1 years) treated with rivaroxaban for a mean duration of six months. Modified Rankin scale (mRS) at admission averaged 2.3 and significantly improved to 0.35 at discharge, with a 95% excellent outcome rate at the end of follow-up. Complete recanalization was noted in 60% of patients without any major bleeding complications, further adding to the evidence for rivaroxaban as a primary anticoagulation option in clinically stable CVT patients [32]. Similar findings were observed in a Polish group of 36 CVT patients treated with DOACs [18] with Dabigatran, 10 with Rivaroxaban, and 8 with Apixaban), with 94.4% of patients showing partial recanalization and 55.6% showing complete recanalization after a follow-up of more than 8 months. Favorable functional outcome defined as improvement in modified Ranking score was seen in up to 70% of patients, with 10% experiencing major bleeding episodes (menorrhagia and gastrointestinal) [33]. The recurrence rate of CVT noted in this study is about 5%, comparable to those major studies that used VKAs like warfarin [34]. A multicenter cohort study of 111 patients (VKAs (66), Rivaroxaban (36), Dabigatran (9)) found excellent functional outcomes (defined by mRS of 0–2) in the DOAC group (90% versus 79%) compared to warfarin group with similar rates of recanalization, recurrent CVT, ICH and non-ICH bleeds in either group over a median period of 8 months. Despite limitations like open-label design, lack of randomization, and adequate radiological data for follow-up, the findings of the study provide external validity for the use of NOACs in CVT patients [35]. Additionally, small comparative observational studies have shown similar outcomes in both efficacy and safety parameters between different DOAC agents [36,37]. Another retrospective age- and sex-matched cohort study of 27 CVT patients receiving DOACs, showed a recanalization rate of 70% at a median follow-up of six months with a median mRS of 0 at follow-up without any significant difference in bleeding tendencies between the groups [38]. The generalizability of these prospective studies is limited by the small sample size, exclusion of critically ill and under-representation of females and postpartum CVT, and lack of a control group with standard-of-care comparator (VKAs/Heparin). ACTION-CVT One of the largest multicenter retrospective cohort studies analyzed 845 CVT patients (mean age 44.8 years, 64.7% women) across three countries, of which 33.0% received only direct oral anticoagulants (DOACs), 51.8% received only warfarin, and 15.1% received both at different times. During a median follow-up of 345 (IQR, 140–720) days, there were 5.68 recurrent venous thrombosis, 3.77 major hemorrhages, and 1.84 deaths per 100 patient-years. Among 525 patients who met recanalization analysis inclusion criteria, 36.6% had complete, 48.2% had partial, and 15.2% had no recanalization. This study concluded that when compared with warfarin, DOAC treatment was associated with similar risk of recurrent venous thrombosis (HR, 0.94 [95% CI, 0.51–1.73]; *p* = 0.84), death (HR 0.78 [95% CI, 0.22–2.76]; *p* = 0.70), and rate of partial/complete recanalization (OR 0.92 [95% CI, 0.48–1.73]; *p* = 0.79), but a lower risk of major hemorrhage (HR, 0.35 [95% CI, 0.15–0.82]; *p* = 0.02). Limitations include treatment-by-indication bias, 15.7% loss to follow-up within 90 days, low event rate limiting subgroup analyses, inability to distinguish between de novo CVT and CVT extension, lack of INR data in warfarin-treated patients, likely low ascertainment of asymptomatic hemorrhage, and heterogeneous timing of follow-up imaging [39].

### 4.2. Clinical Trials of DOACs in CVT

#### 4.2.1. RE-SPECT Trial

The landmark trial randomized 120 stable CVT patients (after pre-treatment with parenteral anticoagulant of heparin group for 5–15 days) to receive either Dabigatran 150 mg twice a day or dose-adjusted warfarin with a targeted INR of 2 to 3. Rates of recanalization (60% in dabigatran group vs. 67% in warfarin group) and recurrent thromboembolic events were comparable in both groups. Rates of bleeding events also did not differ significantly between the groups. The study excluded specific subsets of CVT (infection, pregnancy, malignancy, and trauma-related CVT). It was underpowered to detect significant differences in recurrent VTE. Although this randomized multicenter study failed to demonstrate the noninferiority or superiority of either drug due to the limited sample size, it established the safety and efficacy of dabigatran and dose-adjusted warfarin in preventing recurrent VTEs in patients with CVT [21].

#### 4.2.2. SECRET Trial

This was a phase II, multicentric open-label, blinded-endpoint randomized trial to assess the feasibility of recruitment, the safety of rivaroxaban compared with standard-of-care anticoagulation, and patient-centered functional outcomes. 55 CVT patients, within 14 days of a new diagnosis of symptomatic CVT, were randomized to receive rivaroxaban 20 mg daily or standard-of-care anticoagulation (warfarin, target international normalized ratio, 2.0–3.0, or low-molecular-weight heparin) for 180 days, with optional extension up to 365 days. There were numerically higher bleeding rates with rivaroxaban compared to comparator control (Warfarin/LMWH), but these rates did not exceed what has been reported in the previous studies, which may be attributed to the unique dosing strategy of 20 mg once a day. Both groups showed similar rates of functional improvement by day 365 (82.6% in rivaroxaban and 84% in controls). SECRET trial is also distinct from RE-SPECT CVT in that lead-in parenteral anticoagulation was not required. The open-label study was underpowered to demonstrate superiority or non-inferiority of DOAC or detect differences in the bleeding rates between treatment groups. Pregnant women, antiphospholipid syndrome, and high-risk populations (trauma, malignancy, and infection related) CVT patients were excluded. However, this trial demonstrated the feasibility of recruitment, opening avenues for conducting adequately powered large-scale multicenter trials exploring DOACs in CVT [40].

#### 4.2.3. CHOICE CVT Trial

A single-center, open-label study randomized 89 Chinese patients with CVT to either dabigatran or warfarin (1:1) transitioned after 10–15 days of LMWH. The primary efficacy and safety endpoints included the number of patients with recurrent CVT and/or deep venous thrombosis (DVT) and major clinical bleeding within 180 days. A statistically insignificant but clinically greater number of patients were found to have recurrent CVT and/or DVT in the dabigatran arm compared to the warfarin group. Recanalization rates and number of bleeding events were similar in both groups, adding to the evidence supporting the role of DOACs in CVT. However, the single center and open-label design of the study limit generalizability to other ethnicities and regions [41]. 

#### 4.2.4. Systematic Review and Meta-Analysis

Several systematic reviews have come up with meta-analyses in this area to further strengthen the evidence and provide a road towards implementation in guideline-directed medical therapy among CVT patients. The clinical characteristics and outcomes of all the patients in various systematic reviews are reviewed in Table 2.

## 5. Consideration of DOACs in Special Populations

Certain notable limitations in the studies above justify the lack of clear consensus guidelines on the usage of DOACs in CVT. These include heterogeneity in study designs, choice of DOAC, bridging strategies and duration of anticoagulation, and exclusion or under-representation of specific risk groups (pregnancy, antiphospholipid antibody syndrome, chronic liver disease, chronic renal disease, active malignancy) [50]. Pregnancy induces a hypercoagulable state that increases the risk of CVT during the postpartum period by up to tenfold [51]. Guidelines recommend against DOACs in pregnant women due to possible crossing of placenta, and fetal toxicity demonstrated in animal models. They are contraindicated in breastfeeding women owing to inadequate safety data with preference given to LMWH [52]. DOACs were not found to be effective in APS and increased the risk of thrombosis compared to VKAs [53,54]. Periodic monitoring of renal function with appropriate dose adjustment is necessary in patients with chronic kidney disease receiving DOACs, especially in the elderly and those on Dabigatran (contraindicated in CrCl < 30 mL/min). The hepatic metabolism of DOACs varies from highest for apixaban and lowest for dabigatran. Consequently, apixaban is contraindicated in Child-Pugh class B and C whereas all of them are contraindicated in Child-Pugh C. Caution with dose adjustment is advised for use in Child-Pugh B for other agents [55]. Drug-drug interactions are, in general, less frequently observed in DOACs compared to VKAs; nevertheless, drugs strongly interacting with P-glycoprotein and CYP3A4/5 (anti-epileptics) can adversely affect the efficacy and/or safety when administered concomitantly with DOACs, particularly in those with deranged renal/hepatic metabolism [56,57].

Most of the data on usage of DOACs is from either patients with deep venous thrombosis or non-valvular atrial fibrillation. Finally, consideration of Frailty and adverse outcomes especially in the geriatric population, the need for a personalized approach, and the evidence of those DOACs with fewer complications extrapolating the data from non-valvular atrial fibrillation, indeed conjugation between an integrated approach to patient management and evaluation of frailty and comorbidities could probably provide an adequate way to evaluate, characterize and stratify risk in anticoagulant therapy. Taking adequate consideration of all clinical characteristics and physiological reserves could aid the physicians in choosing the right OAC drug, either a DOAC or VKA, minimize the risk of adverse events, and optimize the reduction of thromboembolic and death events [58,59,60].

## 6. Choice of DOAC

Evidence on head-to-head comparison within DOACs in CVT is scarce and therefore, no agent is recommended by guidelines. RE-SPECT trial compared the efficacy of dabigatran to warfarin in patients with CVT and found both medications equally effective in preventing recurrent thromboembolism. Similarly, other retrospective and prospective trials have found similar benefits from other DOACs. However, while the effectiveness and safety of DOACs in the general population with atrial fibrillation (AF) are undeniable, the available data suggest that treating frail patients affected by AF with apixaban could guarantee significantly better efficacy and safety than warfarin, also because of relatively greater availability of data relating to geriatric subgroups, impaired renal function subgroups [61,62]. The best way forward could be considering patient-related comorbid risk factors, and the pharmacodynamics of each anticoagulant extrapolating the data from other disease categories must be carefully considered during treatment selection.

## 7. Future Prospectus of DOACs in CVT

### RWCVT Trial

This is a phase 2 randomized control trial comparing outcomes in CVT patients receiving rivaroxaban versus warfarin. The primary outcome is sinus venous thrombosis severity scale (SVTSS) at 6 months, while the secondary outcomes are functional status, raised intracranial pressure, and bleeding events. These trials could address current knowledge gaps and enhance the evidence base to establish new guideline-directed medical therapy (GDMT) for the management of patients with CVT. While most of the information regarding the usage of DOACs has been derived from case series, case reports, and prospective and retrospective cohort studies, RCTs supporting the same are few. Patients enrolled in various studies were followed much more closely than in real-world scenarios and, hence, may not represent the general population. Although there has been comparable efficacy and safety to the standard of care demonstrated in various studies using DOACs, the occurrence of any major bleeding event may warrant reversal agents. The reversal agents for the treatment of DOAC-related life-threatening bleeding are limited due to cost and availability compared to reversal agents used in VKA toxicity.

The ongoing clinical trials (Table 3), such as the above RWCVT trial, highlight the promising future of research in this area. However, there is currently a lack of sufficient randomized controlled trials (RCTs), and larger, long-term RCTs are needed to strengthen the evidence for the use of DOACs. Additionally, there is limited knowledge regarding the use of DOACs in real-world clinical settings, so further research is necessary to establish detailed eligibility criteria based on patient characteristics (Table 4).

## 8. Conclusions

The literature presents compelling evidence regarding the efficacy and safety of direct oral anticoagulants (DOACs) in the management of cerebral venous sinus thrombosis (CVT). Despite some limitations, such as small sample sizes and the absence of large randomized controlled trials, the existing evidence supports the consideration of DOACs as a viable option for the treatment of CVT in terms of high rates of recanalization, low recurrence rates, and minimal bleeding complications. Moreover, ongoing research will further contribute to strengthening the evidence base and guiding the implementation of DOACs in guideline-directed medical therapy for CVT in the future.However, given the scarcity of data, real-world registry-based studies, and/or retrospective studies are an unmet need and would further contribute to strengthening the evidence base and guiding the implementation of DOACs in guideline-directed medical therapy for CVT in the future.

## Figures and Tables

**Table 1 biomedicines-13-00189-t001:** Comparison of different guidelines for the management of cerebral venous thrombosis.

	AHA/ASA Guidelines [5]	ESO Guidelines [13]
Year of publication	2011	2017
Diagnostic investigation	MRI with MRV or CT with CTV	MRI with MRV or CT with CTV
Acute medical treatment	IV/SC heparin	IV/SC Heparin
Long term anticoagulation	VKAs like warfarin	VKAs like warfarin
Duration of anticoagulation for provoked causes	3–6 months	3–6 months
Duration of anticoagulation for unprovoked causes	6–12 months	6–12 months
Duration for inherited Thrombophilia	Life long after first recurrence	Life long after first recurrence
Role of mannitol	Not recommended	Not recommended
Role of Acetazolamide	Recommended	Not recommended
Role of Corticosteroids	Not recommended	Not recommended
Surgical management	Severe mass effect/Clinical deterioration consider surgical management	Severe mass effect/Clinical deterioration consider surgical management
Seizure prophylaxis	Not recommended	Not recommended
Follow up Imaging	3–6 months	3–6 months
Role of DOACs	Currently Not recommended	Currently Not recommended

**Table 2 biomedicines-13-00189-t002:** Summary of data on published Systematic Review and Meta-analyses.

Author	Year	Category	DOACs	Number of Studies (*n*)	Main Outcomes
Sheng et al. [42]	2020	Systematic review	64% Rivaroxaban 11% Apixaban 25%Dabigatran	11	Favorable mRS (0–1) at 6 months in more than 86.7%. No recurrent VTE at 12 months Recanalization rates at 6 months 55–100%
Lee GKH et al. [43]	2020	Systematic review	38.4% Rivaroxaban 5.65% Apixaban 47%Dabigatran	6	Comparable rates of partial or complete recanalization (RR: 1.02; 95% CI: 0.89–1.16) and functional recovery (RR: 1.02; 95% CI: 0.93–1.13) Lesser major bleeding events in DOACs (RR: 0.44; 95% CI: 0.12–1.59)
Li H et al. [44]	2020	Systematic review	N/A	6	No significant difference between DOACs and VKAs in recurrence of VTE or death (RR: 0.34; 95% CI: 0.06–1.98), partial recanalization (RR: 0.97; 95% CI: 0.93–1.14), and overall hemorrhage events (RR: 0.86; 95% CI: 0.47–1.58)
Nepal G et al. [45]	2021	Metanalysis	Dabigatran Rivaroxaban Apixaban	17	No difference in efficacy and safety. Complete recanalization more likely in DOAC group (RR: 1.33; 95% CI: 0.98–1.82; I^2^: 73.13%; *p* = 0.41)
Bose G et al. [46]	2021	Systematic review	41.2% Dabigatran, 47.3% Rivaroxaban, 9.7% Apixaban and 1.8% Edoxaban	33	Similar risk of death in both the arms (RR: 2.12; 95% CI: 0.29–15.59) Favorable mRS (0–2) in DOACs category. (RR: 1.13; 95% CI: 1.02–1.25) New ICH in 1.7% and recurrent CVT in 1.5% in DOAC treated patients
Yaghi S et al. [47]	2022	Systematic review and metanalysis	38.4% (Rivaroxaban, Apixaban, Dabigatran and Edoxaban)	19	Direct oral anticoagulants (DOACs) exhibited similar risks for recurrent venous thromboembolism (RR: 0.85; 95% CI: 0.52–1.37; I^2^ = 0%), major bleeding events (RR: 0.70; 95% CI: 0.40–1.20; I^2^ = 0%), and the rate of complete venous recanalization (RR: 0.98; 95% CI: 0.87–1.11; I^2^ = 0%)
Riva N et al. [48]	2022	Systematic review and metanalysis	49% rivaroxaban, 27.8% dabigatran, 18.6% apixaban and 4% not specified	23	There was no notable variation in mortality rate (1.76%, 95% CI: 0.70–3.24%; I^2^ = 0%); major bleeding events occurred at a rate of 2.41% (95% CI: 1.26–3.91%; I^2^ = 1.5%); recurrent thrombosis was observed at 2.05% (95% CI: 1.04–3.37%; I^2^ = 0%); excellent neurological outcomes were achieved in 85.9% of cases (95% CI: 79.0–91.7%; I^2^ = 63.7%); and vessel recanalization was reported at 89.0% (95% CI: 82.9–93.9%; I^2^ = 62.7%)
Ranjan et al. [49]	2024	Metanalysis	2301 patients (Dabigatran, Apixaban, Edoxaban, Rivaroxaban and warfarin)	25	Favorable long-term mRS scores of 0–2 (risk ratio [RR] = 1.01, 95% CI = 0.98–1.03; *p* = 0.61), incidence of new intracranial hemorrhage (RR = 1.00, 95% CI = 0.48–2.08; *p* = 0.99), all-cause mortality (RR = 1.00, 95% CI = 0.50–1.98; *p* = 0.99), absence of recanalization (RR = 0.95, 95% CI = 0.77–1.18; *p* = 0.65), and recurrent venous thrombosis events (RR = 0.63, 95% CI = 0.33–1.22; *p* = 0.17) were found to be comparable between the two treatment groups. However, subgroup analysis revealed a lower recurrence rate of venous thrombosis in the rivaroxaban group compared to the warfarin group (2.2% vs. 8.5%, RR = 0.33, 95% CI = 0.11–0.98; *p* = 0.05).

**Table 3 biomedicines-13-00189-t003:** Ongoing Studies for Cerebral Venous Sinus Thrombosis.

Author	Type of Study	Number of Patients	Outcomes
Munckof et al. [63]	Prospective cohort study	500	The main outcome measure is a combined endpoint consisting of recurrent venous thrombosis and significant bleeding events evaluated at the 6-month follow-up. Additionally, an adjusted odds ratio for the primary endpoint is determined using propensity score inverse probability treatment weighting.

**Table 4 biomedicines-13-00189-t004:** Summary of data on DOAC in CVT.

Name of Study	Type of Study	Number of Patients	Comparative Arm	Outcomes
Yaghi et al. [39] (ACTION-CVT)	Retrospective study	845 patients	DOAC vs. warfarin	DOACs was associated with similar clinical and radiographic outcomes and favorable safety profile when compared with warfarin treatment.
Esmaili et al. [64]	Prospective cohort study	36 patients	Rivaroxaban vs. warfarin	In total, 13 patients (36.11%) were treated with Warfarin, while 23 patients (63.89%) received Rivaroxaban. An optimal mRS score (0–1) was achieved in 9 out of 10 patients (90%) in the Rivaroxaban group and in 19 out of 22 patients (86.36%) in the Warfarin group. Magnetic resonance venography (MRV) revealed complete or partial recanalization in 12 of 14 patients (85.71%) treated with Rivaroxaban, compared to all patients in the Warfarin group. No significant differences were observed between the two groups concerning the occurrence of major or minor hemorrhages.
Ferro JM et al. [21] (RE-SPECT CVT)	Open-label randomized clinical trial	120 patients	Dabigatran vs. warfarin	This trial found that anticoagulation with either dabigatran or warfarin had a low risk of recurrent VTEs, and the risk of bleeding was similar in both arms.
Maqsood et al. [65]	Prospective study	45 patients	Rivaroxaban vs. warfarin	Of the 45 patients, overall recanalization was observed in 18 cases (86%) in the rivaroxaban group and 20 cases (83%) in the warfarin group at the six-month follow-up, with all 45 cases (100%) in both groups achieving recanalization by the 12-month follow-up. An excellent outcome, defined as an NIHSS score of 0, was achieved by 20 patients (95%) in the rivaroxaban group at three- to 12-month follow-ups and by 23 patients (96%) at six- to 12-month follow-ups. No major bleeding events or thrombotic recurrences were reported.
Bajko et al. [66]	Retrospective study	87 patients	VKAs vs. DOAC (Rivaroxaban and Apixaban)	Out of 87 patients, 7 received DOAC (6 Rivaroxaban and 1 Apixaban). 5/6 patients had data at 6 months. Excellent outcome was achieved in 4/5 patients with no hemorrhagic events or recurrent thrombosis.
Jiang et al. [67]	Retrospective study	83 patients	Rivaroxaban vs. warfarin	The primary outcomes were a composite of recurrent thrombosis or bleeding outcomes. There was no significant difference. Indeed, the median duration to recovery time was shorter in the DOAC group compared to warfarin group.
Ma et al. [41]	Open label RCT	89 patients	Dabigatran vs. warfarin	At day 180, the risk of recurrent venous thrombosis was not statistically significant (8 vs. 3; *p* = 0.09) and no difference in overall major bleeding rates and CSNMB. Recanalization rates were comparable between both the groups.
Fatima et al. [68]	Prospective study	31 patients	Rivaroxaban, no comparator arm	Excellent functional outcome was achieved in 93% of patients with a mortality of 6%. Recurrent CVT was developed in 3% of patients at 6 months of follow-up.
Hsu et al. [37]	Retrospective study	46 patients	Warfarin vs. DOACs	No significant difference in efficacy and safety outcomes.
Lurkin et al. [69]	Retrospective study	41 patients	Warfarin vs. DOAC	No recurrent thrombosis in the DOAC group vs. 3 patients in the warfarin group experienced recurrence. No bleeding events in either group. 40% complete recanalization vs. 28.6% in the warfarin group.
Christodoulides et al. [70]	Retrospective study	49 patients	Warfarin vs. Factor Xa inhibitors	Higher rates of recanalization in the DOAC group as compared to the warfarin group within 12 months (69.2% vs. 33.3%, *p* = 0.054). No statistically significant difference in bleeding-related events was observed.
Shahid et al. [71]	Retrospective study	36 patients	Warfarin vs. DOAC	There was no statistically significant difference in recanalization status, although it proved to be better tolerated, as none of the patients stopped the treatment due to adverse events and the risk of major bleeding was significantly low in the NOAC group. Nine patients in the warfarin group stopped medication, while none in the NOAC group did so (*p* = 0.034).
Powell et al. [36]	Retrospective study	119 patients	Warfarin vs. Factor Xa inhibitors vs. Heparin	The analysis included a total of 119 patients divided into three treatment groups: warfarin (89 patients), enoxaparin (11 patients), and oral factor Xa inhibitors (19 patients). The incidence of recurrent thrombotic events was 11.2% in the warfarin group, 0% in the enoxaparin group, and 10.5% in the oral factor Xa inhibitor group (*p* = 0.7635). No significant differences were observed between the groups for secondary outcomes, including recanalization, bleeding events, or functional status.
Mendonca et al. [72]	Retrospective study	18 patients	Warfarin vs. Dabigatran	A cohort of 15 patients received treatment with dabigatran, with a median follow-up duration of 19 months. Among these patients, 87% achieved excellent outcomes, and recanalization was observed in 80% of cases within the dabigatran group.

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
