# Peer review of "Direct-Acting Oral Anticoagulants in the Management of Cerebral Venous Sinus Thrombosis—Where Do We Stand?"

_biomedicines, 2025, doi:10.3390/biomedicines13010189_

Round 1
Reviewer 1 Report
Comments and Suggestions for Authors
Dear authors,
1. The search strategy is detailed, with the use of multiple databases such as PubMed, EMBASE, and Google Scholar to retrieve 1,211 articles. However, it would be helpful if the specific search queries (e.g., details of the AND/OR combinations) were more clearly stated, as this would allow readers to replicate the search more easily. In particular, explaining the process used to narrow down the search results (such as inclusion and exclusion criteria) would enhance the transparency and reliability of the review.
2. The reasons for transitioning from VKAs (vitamin K antagonists) to DOACs, such as improving patient adherence, reducing dietary restrictions, and minimizing the burden of monitoring, are emphasized. However, a more detailed discussion on the specific conditions or clinical situations where DOACs are recommended would be beneficial. In particular, providing clear guidelines on the patient populations for whom DOACs are most useful (e.g., those with low bleeding risk or patients for whom regular monitoring is difficult) would be helpful for readers.
3. The review includes topics related to COVID-19 pandemic and post-vaccination thrombosis, suggesting that DOACs may be effective in these conditions. However, including more detailed evidence or research findings on the role of DOACs in emerging conditions such as vaccine-induced thrombotic thrombocytopenia (VITT) would help readers better understand their clinical applicability.
4. The scope of DOACs' applicability in clinical practice is not clearly defined, particularly in the management of CVT patients. It is not discussed how DOACs might be beneficial or which specific patient populations would benefit the most. Future research should provide more concrete guidelines on the clinical situations where DOACs are recommended, such as for patients with low bleeding risk or those for whom monitoring is difficult.
5. It is mentioned that there is a lack of direct comparison studies between different DOACs. While the RE-SPECT trial compared dabigatran with warfarin, data comparing other DOACs (such as rivaroxaban and apixaban) are limited. As a result, guidelines for drug selection remain unclear. Further discussion is needed on which DOAC is most suitable based on patients' comorbidities and the pharmacological characteristics of the drugs.
6. The ongoing clinical trials, such as the RWCVT trial, highlight the promising future of research in this area. However, there is currently a lack of sufficient randomized controlled trials (RCTs), and larger, long-term RCTs are needed to strengthen the evidence for the use of DOACs. Additionally, there is limited knowledge regarding the use of DOACs in real-world clinical settings, so further research is necessary to establish detailed eligibility criteria based on patient characteristics.
Author Response
Manuscript ID: Biomedicines-3364122
Title: DIRECT-ACTING ORAL ANTICOAGULANTS IN THE MANAGEMENT OF CEREBRAL
VENOUS SINUS THROMBOSIS – WHERE DO WE STAND?
Reviewer Comments to Authors:
Reviewer 1:
The search strategy is detailed, with the use of multiple databases such as PubMed,
EMBASE, and Google Scholar to retrieve 1,211 articles. However, it would be helpful if the
specific search queries (e.g., details of the AND/OR combinations) were more clearly stated,
as this would allow readers to replicate the search more easily. Explaining the process used
to narrow down the search results (such as inclusion and exclusion criteria) would enhance
the transparency and reliability of the review.
Authors response: We greatly appreciate the reviewers’ constructive comment. The
following concerns are addressed in the Search strategy section. The modified text is as
follows:
Search Strategy: This narrative review focused on an overview of the usage of direct oral
anticoagulants in patients with cerebral venous sinus thrombosis. We searched PubMed
(MEDLINE and PubMed Central), EMBASE, and GOOGLE SCHOLAR, using the search
terms “Cerebral Venous Sinus Thrombosis”, “Cerebral Venous Thrombosis”, “CVT”,
“CVST”, “CSVT” and retrieved 1211 results. Limiting our search to CVT (OR) CSVT (OR)
CSVT (OR) Cerebral Venous Sinus Thrombosis” (OR) Cerebral Venous Thrombosis (AND)
Dabigatran (OR), Rivaroxaban (OR), Apixaban (OR), DOAC (OR) NOAC (OR) Non-VKA
anticoagulants and filtering all duplicates from all databases, we included all the available
literature or studies till November 30, 2024. The inclusion criteria used are: 1) Studies
describing the adult population 2) All types of studies like cohort studies (Both Prospective
and retrospective cohort studies), comparative studies, Randomized studies, and systematic
reviews and meta-analyses are included. The exclusion criteria applied during the search
were: 1) Case reports and case series, 2) Studies reporting data on pregnant patients 3)
Studies reporting data only on children. Additional articles of critical importance were
included based on the author’s knowledge.
The reasons for transitioning from VKAs (vitamin K antagonists) to DOACs, such as
improving patient adherence, reducing dietary restrictions, and minimizing the burden of
monitoring, are emphasized. However, a more detailed discussion on the specific conditions
or clinical situations where DOACs are recommended would be beneficial. Providing clear
guidelines on the patient populations for whom DOACs are most useful (e.g., those with low
bleeding risk or patients for whom regular monitoring is difficult) would be helpful for readers.
Authors response: We greatly appreciate the reviewers’ constructive comment. The same
has been added under the “Why there is a need for paradigm shift section”. The modified
text is as follows: Recently there has been increased usage of DOACs for various indications
in the elderly population as well. Octogenarians and nonagenarians are not an exception. In a
meta-analysis comprised of 547419 elderly patients with atrial fibrillation, it was found that
Compared with VKAs, DOACs significantly reduced risk for stroke (OSs, HR: 0.87, 95% CI:
0.81–0.94; RCT, RR: 0.82, 95% CI: 0.67–0.96), and Intracranial hemorrhage (OSs: 0.47
[0.37–0.57]; RCTs: 0.47 [0.31–0.63]), without increasing risk for Gastrointestinal Bleeding
(GIB) (OSs: 1.21 [0.98–1.43]; RCTs: 1.34 [0.91–1.77]), and all-cause mortality (OSs: 1.01
[0.92–1.11]; RCTs: 0.94 [0.87–1.00]). Among OSs, DOACs significantly decreased the risk
for major bleeding (0.87 [0.77–0.98]) and MI (0.89 [0.79–0.99]). However, it was found that
dabigatran, but not other DOACs, significantly increased the risk for GIB (1.48 [1.23–1.72]).
So, it shows that DOACs are also safe in patients who have high bleeding risk compared to
VKAs, and they indeed reduce the risk of major bleeding diathesis.
The review includes topics related to the COVID-19 pandemic and post-
vaccination thrombosis, suggesting that DOACs may be effective in these
conditions. However, including more detailed evidence or research findings on
the role of DOACs in emerging conditions such as vaccine-induced thrombotic
thrombocytopenia (VITT) would help readers better understand their clinical
applicability.
Authors response: We greatly appreciate the reviewers’ constructive comment. The same
has been added.. The modified text is as follows: VITT remains relevant in low—and
middle-income countries (LMICs) that can only afford adenoviral vector-based vaccines and
where the vaccination campaign is ongoing. Understanding which vaccine constituent(s)
trigger(s) the immune response to PF4 is also important for designing safer delivery
systems. This appears to be a class effect of all adenoviral vector vaccines. Treatment of
VIITT includes intravenous immunoglobulin (IVIG) which inhibits FCRY2A mediated
platelet activation and non-heparin anticoagulants like Direct Thrombin Inhibitors, Factor Xa
inhibitors, and Fondaparinux. VKAs are avoided in the acute thrombotic phase, as protein C
deficiency induced by VKAs might provoke extensive thrombotic phenomena.
The scope of DOACs' applicability in clinical practice is not clearly defined,
particularly in the management of CVT patients. It is not discussed how
DOACs might be beneficial or which specific patient populations would benefit
the most. Future research should provide more concrete guidelines on the
clinical situations where DOACs are recommended, such as for patients with
low bleeding risk or those for whom monitoring is difficult.
Authors response: We greatly appreciate the reviewers’ constructive comment. The same
has been added under the special considerations section.
It is mentioned that there is a lack of direct comparison studies between
different DOACs. While the RE-SPECT trial compared dabigatran with
warfarin, data comparing other DOACs (such as rivaroxaban and apixaban)
are limited. As a result, guidelines for drug selection remain unclear. Further
discussion is needed on which DOAC is most suitable based on patients'
comorbidities and the pharmacological characteristics of the drugs.
Authors response: We greatly appreciate the reviewers’ constructive comment. The same
has been added to the choice of DOACs section.
The ongoing clinical trials, such as the RWCVT trial, highlight the promising
future of research in this area. However, there is currently a lack of sufficient
randomized controlled trials (RCTs), and larger, long-term RCTs are needed to
strengthen the evidence for the use of DOACs. Additionally, there is limited
knowledge regarding the use of DOACs in real-world clinical settings, so
further research is necessary to establish detailed eligibility criteria based on
patient characteristics.
Authors response: We greatly appreciate the reviewers’ constructive comment. The same
has been added in the Ongoing trials section.
Reviewer 2:
This manuscript is a narrative review of the usage of direct oral anticoagulants in patients with
cerebral venous sinus thrombosis (CVT). The paper is well-written and easy to read.
Authors response: We greatly appreciate the reviewers’ constructive comment
The data about the issue are still scarce, real-world registry and/or retrospective studies
should be encouraged in the conclusions.
Authors response: We greatly appreciate the reviewers’ constructive comment. The
following changes have been added to the conclusions section. The modified text appears as
follows: … However, given the scarcity of data, real-world registry-based studies, and/or
retrospective studies are an unmet need and would further contribute to strengthening the
evidence base and guiding the implementation of DOACs in guideline-directed medical
therapy for CVT in the future.
Many data are available about DOACs in deep vein thrombosis and non-valvular AF. Thus,
some considerations might be translated to CVT, still considering the differences of the main
problem. For example, regarding special populations and choice of DOACs, I suggest
considering the issue of frailty and adverse outcomes, the need for a personalized approach,
and the evidence of those DOACs with fewer complications in CVT as it is for NVAF, indeed
the conjunction between an integrated approach to patients’ management and evaluation of
frailty and comorbidities could probably provide an adequate way to evaluate, characterize
and stratify risk in anticoagulant therapy. Taking adequate consideration of all clinical
characteristics and physiological reserves could aid the physicians to choose the right OAC
drug, either a DOAC or VKA, minimize the risk of adverse events and optimize the reduction
of thromboembolic and death events. See Proietti M et al doi: 10.1016/j.arr.2022.101652 and
doi: 10.3390/jpm12030469. PMID: 35330468; Lucà et al doi: 10.3390/jcm12185955.
Authors response: We greatly appreciate the reviewers’ constructive comment. The
modified text under the special consideration section is as follows: … Most of the data on
usage of DOACs is from either patients with deep venous thrombosis or non-valvular atrial
fibrillation. Finally, consideration of Frailty and adverse outcomes especially in the geriatric
population, the need for a personalized approach, and the evidence of those DOACs with
fewer complications extrapolating the data from non-valvular atrial fibrillation, indeed
conjugation between an integrated approach to patient management and evaluation of frailty
and comorbidities could probably provide an adequate way to evaluate, characterize and
stratify risk in anticoagulant therapy. Taking sufficient consideration of all clinical
characteristics and physiological reserves could aid the physicians in choosing the right OAC
drug, either a DOAC or VKA, minimize the risk of adverse events, and optimize the
reduction of thromboembolic and death events.
Similarly, regarding the choice of a specific DOAC, while the effectiveness and safety of
DOACs in the general AF population is undeniable, the available data suggest that treating
frail patients affected by AF with apixaban could guarantee significantly better efficacy and
safety than warfarin, also because of relatively greater availability of data relating to geriatric
subgroups, impaired renal function subgroups. (Grymonprez M, et al doi:
10.3389/fphar.2020.583311. Monelli et al. doi: 10.2147/VHRM.S191208. PMID: 30833810;
PMCID: PMC6378887.
Authors response: We greatly appreciate the reviewers’ constructive comment. The
modified text under the Choice of DOAC section is as follows: …However, while the
effectiveness and safety of DOACs in the general population with atrial fibrillation (AF) are
undeniable, the available data suggest that treating frail patients affected by AF with apixaban
could guarantee significantly better efficacy and safety than warfarin, also because of
relatively greater availability of data relating to geriatric subgroups, impaired renal function
subgroups. The best way forward could be to consider patient-related comorbid risk factors,
and the pharmacodynamics of each anticoagulant, extrapolating the data from other disease
categories, which must be carefully considered during treatment selection.
We have expanded the description of the search strategy, modified the text according to
reviewers' comments, and changed the conclusion section. We believe these revisions
address your concerns and enhance the clarity and depth of our manuscript. We are grateful for the opportunity to improve our work based on your valuable feedback.
Sincerely,
Authors

Reviewer 2 Report
Comments and Suggestions for Authors
This manuscript is a narrative review on usage of direct oral anticoagulants in patients with cerebral venous sinus thrombosis (CVT). The paper is well written and easy to read.
The following comments have the aim to improve the manuscript
- The data about the issue are still scarce, real-world registry and/or retrospective studies should be encouraged in the conclusions.
- Many data are available about DOACs in deep vein thrombosis and non valvular AF. Thus, some considerations might be translated to CVT, still considering the differences of the main problem. For example, regarding special populations and choice of DOACs, I suggest to consider the issue of frialty and adverse outcomes, the need for a personalized approach and the evidence of those DOACs with less complications in CVT as it is for NVAF, indeed the conjunction between an integrated approach to patients’ management and evaluation of frailty and comorbidities could probably provide an adequate way to evaluate, characterize and stratify risk in anticoagulat therapy. Taking adequate consideration of all clinical characteristics and physiological reserves could aid the physicians to choose the right OAC drug, either a DOAC or VKA, minimize the risk of adverse events and optimize the reduction of thromboembolic and death events. See Proietti M et al doi: 10.1016/j.arr.2022.101652 and doi: 10.3390/jpm12030469. PMID: 35330468; Lucà et al doi: 10.3390/jcm12185955.
- Similarly, regarding the choice of a specific DOAC, while the effectiveness and safety of DOACs in the general AF population is undeniable, the available data suggest that treating frail patients affected by AF with apixaban could guarantee significantly better efficacy and safety than warfarin, also because of relatively greater availability of data relating to geriatric subgroups, imaired renal function subgroups. (Grymonprez M, et al doi: 10.3389/fphar.2020.583311. Monelli et al. doi: 10.2147/VHRM.S191208. PMID: 30833810; PMCID: PMC6378887.
MINORS:
1) in introdution first abbreviation CVT must be defined as Cerebral venous sinus thrombosis
2) check and match text and tables, tables 3 is not a new table but part of table 2
Author Response
-
Reviewer 2:
This manuscript is a narrative review of the usage of direct oral anticoagulants in patients with
cerebral venous sinus thrombosis (CVT). The paper is well-written and easy to read.
Authors response: We greatly appreciate the reviewers’ constructive comment
The data about the issue are still scarce, real-world registry and/or retrospective studies
should be encouraged in the conclusions.
Authors response: We greatly appreciate the reviewers’ constructive comment. The
following changes have been added to the conclusions section. The modified text appears as
follows: … However, given the scarcity of data, real-world registry-based studies, and/orretrospective studies are an unmet need and would further contribute to strengthening the
evidence base and guiding the implementation of DOACs in guideline-directed medical
therapy for CVT in the future.
Many data are available about DOACs in deep vein thrombosis and non-valvular AF. Thus,
some considerations might be translated to CVT, still considering the differences of the main
problem. For example, regarding special populations and choice of DOACs, I suggest
considering the issue of frailty and adverse outcomes, the need for a personalized approach,
and the evidence of those DOACs with fewer complications in CVT as it is for NVAF, indeed
the conjunction between an integrated approach to patients’ management and evaluation of
frailty and comorbidities could probably provide an adequate way to evaluate, characterize
and stratify risk in anticoagulant therapy. Taking adequate consideration of all clinical
characteristics and physiological reserves could aid the physicians to choose the right OAC
drug, either a DOAC or VKA, minimize the risk of adverse events and optimize the reduction
of thromboembolic and death events. See Proietti M et al doi: 10.1016/j.arr.2022.101652 and
doi: 10.3390/jpm12030469. PMID: 35330468; Lucà et al doi: 10.3390/jcm12185955.
Authors response: We greatly appreciate the reviewers’ constructive comment. The
modified text under the special consideration section is as follows: … Most of the data on
usage of DOACs is from either patients with deep venous thrombosis or non-valvular atrial
fibrillation. Finally, consideration of Frailty and adverse outcomes especially in the geriatric
population, the need for a personalized approach, and the evidence of those DOACs with
fewer complications extrapolating the data from non-valvular atrial fibrillation, indeed
conjugation between an integrated approach to patient management and evaluation of frailty
and comorbidities could probably provide an adequate way to evaluate, characterize and
stratify risk in anticoagulant therapy. Taking sufficient consideration of all clinical
characteristics and physiological reserves could aid the physicians in choosing the right OAC
drug, either a DOAC or VKA, minimize the risk of adverse events, and optimize the
reduction of thromboembolic and death events.
Similarly, regarding the choice of a specific DOAC, while the effectiveness and safety of
DOACs in the general AF population is undeniable, the available data suggest that treating
frail patients affected by AF with apixaban could guarantee significantly better efficacy and
safety than warfarin, also because of relatively greater availability of data relating to geriatric
subgroups, impaired renal function subgroups. (Grymonprez M, et al doi:
10.3389/fphar.2020.583311. Monelli et al. doi: 10.2147/VHRM.S191208. PMID: 30833810;
PMCID: PMC6378887.
Authors response: We greatly appreciate the reviewers’ constructive comment. The
modified text under the Choice of DOAC section is as follows: …However, while the
effectiveness and safety of DOACs in the general population with atrial fibrillation (AF) are
undeniable, the available data suggest that treating frail patients affected by AF with apixaban
could guarantee significantly better efficacy and safety than warfarin, also because of
relatively greater availability of data relating to geriatric subgroups, impaired renal functionsubgroups. The best way forward could be to consider patient-related comorbid risk factors,
and the pharmacodynamics of each anticoagulant, extrapolating the data from other disease
categories, which must be carefully considered during treatment selection.We have expanded the description of the search strategy, modified the text according to
reviewers' comments, and changed the conclusion section. We believe these revisions
address your concerns and enhance the clarity and depth of our manuscript. We are grateful
for the opportunity to improve our work based on your valuable feedback.
Sincerely,
Authors
Round 2
Reviewer 2 Report
Comments and Suggestions for Authors
Thank you